# Ophiostomatales Associated with Mediterranean Pine Engraver, *Orthotomicus erosus* (Coleoptera, Curculionidae) in Dalmatia, Croatia

**DOI:** 10.3390/jof8080788

**Published:** 2022-07-28

**Authors:** Marta Kovač, Daniel Rigling, Milan Pernek

**Affiliations:** 1Croatian Forest Research Institute, Division for Forest Protection and Game Management, Cvjetno Naselje 41, 10450 Jastrebarsko, Croatia; milanp@sumins.hr; 2Swiss Federal Research Institute WSL, 8903 Birmensdorf, Switzerland; daniel.rigling@wsl.ch

**Keywords:** blue-stain fungi, *Pinus halepensis*, *Pinus pinea*, climate change, aridification, culture selective medium, outbreak

## Abstract

Mediterranean pine engraver, *Orthotomicus erosus* was never considered as a significant pest in Croatia and did not appear in high population densities until 2017, when it reached outbreak level in Aleppo pine stands. The beetle was first detected in Marjan Forest Park, Split, and was soon recorded in other parts of the Dalmatian coast. Soon after the outbreak occurred, we observed that all of the attacked trees exhibit severe blue staining in the sapwood which indicated fungal infection caused by the *Ophiostomatales* group of fungi. This raised the need to investigate their relationship with *O. erosus* and the pine decline, and the main aim of this study was to isolate and identify them. Isolates were obtained from adult *O. erosus* beetles, their galleries, and blue-stained sapwood, and identified according to the morphological characteristics and DNA sequencing. A total of six *Ophiostomatales* (*Ophiostoma ips*, *O. piceae*, *Graphilbum* cf. *rectangulosporium*, *O. floccosum*, *Sporothrix pseudoabietina* and *Ceratocystiopsis* cf. *minuta*) were identified in the study. This is the first record of *Ophiostomatales* as organisms associated with the pest *O. erosus* and pine species in Croatia.

## 1. Introduction

Associations between fungi and bark beetles (Coleoptera, Curculionidae, Scolytinae) are well known, and those involving tree-infecting fungi and bark beetles as their insect vectors are most researched and best understood [1,2,3]. In some cases, bark beetles can be primary pests, causing serious dieback of forests in huge areas [4,5,6] and significant economic losses [7,8], and in other cases, they are secondary pests whereby they vector pathogenic fungi that can kill trees [9,10,11]. Phytopathogenic fungi associated with tree-killing bark beetles are critical for overwhelming tree defences and incurring host tree mortality [12]. It is assumed that some fungal species help their vectors break the defensive mechanisms of the host plant. This is a successful strategy for weakening the host plant and plays an important role in the attack strategy of bark beetles [13,14,15]. Many pine scolytids carry blue-stain fungi, several of which have been shown to play an important role in the successful infestation of living trees [16,17,18], while others are mainly saprophytic. Blue-stain fungi cause sapstain, a grey, black, or bluish discoloration of the sapwood caused by the presence of pigmented fungal hyphae in the tracheids [19]. The *Ophiostomatales* are known as blue-stain fungi because of the intensive discoloration of the sapwood after bark beetle attack. They belong to the ascomycete families Ceratocystidaceae and Graphiaceae (order Microascales) and the singly family Ophiostomataceae in the order *Ophiostomatales* [14,20]. Some species can display high levels of virulence to their host trees, cause intensive staining of freshly exposed sapwood, and significantly influence the timber quality by lowering its economic value [20,21].

In Croatia, very little research has been done on blue-stain *Ophiostomatales* fungi associated with bark beetles. Those fungi were investigated only in association with *Pityokteines spinidens* Reitter, a phloem-feeding bark beetle that caused a severe decline of silver fir (*Abies alba* Mill.) in various parts of the country [20,22]. So far, blue-stain fungi have not been studied on any pine-infesting insects, nor in the Mediterranean region in Croatia.

Mediterranean pine engraver, *Orthotomicus erosus* Wollaston is widely distributed across the Mediterranean area (southern Europe, northern Africa) and Asia. The pine beetle has been introduced to Fiji, South Africa, Swaziland [23], and the USA [24]. The dominant host is pine with reported damage on Turkish pine (*Pinus brutia*), Caribbean pine (*P. caribaea*), Canary Island pine (*P. canariensis*), shortleaf pine (*P. echinata*), Afghan pine (*P. elderica*), Aleppo pine (*P. halapensis*), black pine (*P. nigra*), stone pine (*P. pinea*), maritime pine (*P. pinaster*), Monterey pine (*P. radiata*), red pine (*P. resinosa*), and Scots pine (*P. sylvestris*) [25,26,27]. Although it is usually considered as a secondary colonizer of weakened, recently dead or felled trees, in favourable conditions it can reach high abundance levels and cause mortality of healthy trees [28,29], and several outbreaks of this pest have already been recorded [29,30,31,32]. 

In Croatia, *O. erosus* was never considered as a significant pest and did not occur at high population densities until 2017 when it reached outbreak levels in Aleppo pine stands [29,30,31,32,33,34]. It was first detected in the Marjan Forest Park, Split, and was soon recorded in other parts of the Dalmatian coast, rapidly changing its status from endemic to epidemic, probably in response to warmer temperatures and extended drought periods [29]. After the initial outbreaks, the wide availability of possible hosts facilitated the spread of the beetle to other areas, and it was assumed that the shift in low-temperature limits might enable *O. erosus* to continue conquering the European continent [35]. In Europe, *O. erosus* is usually bivoltine but depending on temperature, vegetation period, and food quality it can develop up to seven generations per year [28,29,36,37].

During the outbreak, many field inspections were conducted, and it was observed that all of the attacked trees exhibit severe blue staining in the sapwood, indicating fungal infection. This raised the need to deepen the research and investigate the possible role of these fungi in the pine decline. The main aim of this study was to isolate and identify these fungi and elucidate their relationship with *O. erosus* for the first time in Croatia.

## 2. Materials and Methods

### 2.1. Study Sites and Collection of Samples

Samples were collected in 2017, 2018, and 2019, primarily at Marjan Forest Park, Split (43°30′33″ N, 16°24′38″ E) (Table 1), the site where the *O. erosus* outbreak was first recorded and was most severe. Other sites along the Dalmatian coast were also selected according to the intensity of the pest occurrence (Sukošan 44°02′54.0″ N, 15°19′22.1″ E; Ošjak 42°57′40″ N, 16°41′7″ E; the island of Lokrum, Dubrovnik 42°37′38.7″ N, 18°07′15.9″ E), and sampling was sporadically conducted depending on the frequency of the field surveys (Table 1). The trunk sections and bark of the Aleppo pine (*Pinus halepensis* Mill.) and stone pine (*Pinus pinea* L.) attacked by *O. erosus*, with visible symptoms of blue staining in the sapwood and galleries were collected (Figure 1). For sampling only wilting and drying pine trees clearly attacked by *O. erosus* were collected. On every collection site, another bark beetle species *Tomicus destruens* Woll. could be found but in a very low population. No tree was attacked by both bark beetle species at the same time.

Samples from Marjan were taken to the Laboratory for Entomological Analysis of Croatian Forest Research Institute, Jastrebarsko, distributed into mesh cages, and subjected to controlled conditions (20 ± 2 °C, L:D = 16:8). Trunk sections were sprayed with distilled water across the entire surface once per day in order to maintain bark moisture. Beetles of *O. erosus* were morphologically identified by using a stereomicroscope and the key for European bark beetles [38]. Each adult beetle was collected or taken out from its gallery using sterilized tweezers, placed individually in a sterile Eppendorf tube (Eppendorf, Hamburg, Germany), and stored at 4 °C until fungal isolation. This whole process was performed within two weeks after samples had been collected in order to prevent the colonization by secondary fungi. The isolation of fungi was performed within a few days after the collection of samples.

### 2.2. Isolation of Fungi from Bark Beetles

Three different methods were used to isolate fungi from bark beetles (data about isolates in Table 2). In the first method, bark beetles, some whole and some crushed, after washing in sterile distilled water with one small drop (<10 µL) of Tween 80 detergent were first plated on non-selective MEA (Malt Extract Agar) or PDA (Potato Dextrose Agar), and after fungi development (within a week, depending on fungal species) on selective nutrient medium for *Ophiostoma* spp. (20 g malt extract, 20 g agar, and 1000 mL deionised water, amended with 0.05% cycloheximide and 0.04% streptomycin), as in Zhou et al. [39]. In the second method, the procedure was the same as above, but the bark beetles were smashed and placed directly onto the medium without surface rinsing. In the third method, bark beetles were placed on the surface of the selective agar medium, allowed to walk over the plate for two hours, and then removed. Plates were incubated at 23 ± 1 °C in the dark for two-five weeks, during which they were regularly checked for fungal growth and sporulation. Cultures were purified by transferring hyphal tips from the edges of individual colonies to fresh MEA or PDA medium in order to obtain pure cultures for further species identification.

### 2.3. Isolation of Fungi from Woody Tissues and Galleries

Woody tissue of the trunk sections (data about isolates in Table 2) with visible blue stain on the sapwood was shredded with laboratory scissors into smaller pieces (approx. 0.5 × 0.5 cm). Each piece of sapwood was surface sterilized first in 70% ethanol for 1 min and then in 1% sodium hypochlorite for 1 min. After rinsing in sterile distilled water for 1 min, the wood samples were dried on a sterile paper towel and then placed on a selective nutrient medium. Isolation of fungi from the blue-stained galleries included surface rinsing of galleries pieces (approx. 0.5 × 0.5 cm) with a solution of sterile distilled water containing 0.05% Tween 80, drying with sterile paper towels, and placing them in the same medium. The plates were incubated in the same laboratory conditions as above, observed daily for fungal growth, and pure cultures were obtained using the same method as described before.

### 2.4. Identification of Fungi

Obtained fungal isolates were first grouped according to their colony morphology (Figure 2 and Figure 3), and their initial morphological identification was done in Laboratory for phytopathology analysis of Croatian Forest Research Institute, based on the morphological keys and descriptions by Harrington [40], Suh et al. [41], and Wang et al. [42]. The fruiting structures were mounted on glass slides in distilled water or lactic acid and examined using an Olympus SZX10 stereo microscope (Olympus Co., Tokyo, Japan), and an Olympus BX53 light microscope with differential interference contrast. Measurements were made of 50 of each of the relevant morphological structures so that the ranges and average values of length and width could be calculated. The photographic images were captured with an Olympus XC30 (Tokyo, Japan) digital camera and accompanying software, and images of fungal colonies were taken with Olympus E-30 (Tokyo, Japan) camera. Remaining morphological identifications and molecular analysis of the obtained isolates were performed in the Phytopathology Laboratory of the Swiss Federal Institute for Forest, Snow and Landscape Research (WSL) in Birmensdorf (Switzerland). All fungal cultures were deposited in the Laboratory of Phytopathological Analysis at the Croatian Forest Research Institute (Jastrebarsko, Croatia).

### 2.5. DNA Extraction, PCR, and Sequencing

A total of 60 isolates were included in the molecular studies to confirm the species identification resulting from the morphological examinations. Mycelium of the selected isolates was harvested for DNA extraction, which was then performed using the Thermo Scientific™ KingFisher™ Flex Purification System. The ribosomal ITS region was amplified by PCR using the fungal primers ITS-1 and ITS-4 [43,44] for all of the isolates. In addition, β-tubulin (βT) and elongation factor 1-α (TEF 1-α) were sequenced for 23 isolates that could not be identified to species level by ITS sequencing. Primers used for β-tubulin (βT) were T10 [45] and Bt2b [46], and primers used for TEF 1-α were EF2-F [47], EF2-R [48], F-728F [49] and EF2 [50]. The PCR products were scored on agarose gels, purified, and sequenced in both directions on a 3130xl DNA Analyzer. Sequence editing and analysis were performed by using CLC Main Workbench 7.6.2. and additionally, by using BioEdit 7.0. software. The sequences were blasted in the UNITE, BOLD, and NCBI Genebank sequence database for species identification. The sequences generated for this study are available from GenBank under the accession numbers ON697188-ON697193, ON736861-ON736868 (Appendix A).

## 3. Results

Morphological and molecular analyses of 60 isolates included in the present study revealed six species belonging to Ophiostomatales. Most of the isolates were associated with *Pinus halepensis* as the host tree and Marjan Forest Park (Split) as the most common locality where samples were collected. Overall, 59% of the isolates were recovered from the adult *O. erosus* beetles, and 41% from the blue-stained woody tissues and galleries. The most frequently isolated species was *Ophiostoma ips* (Rumbold) Nannf. (30 out of 60 isolates), which was recovered from both beetles and blue-stained tissue/galleries. 

Isolates identified as *Sporothrix pseudoabietina* H.M. Wang, Q. Lu and Zhen Zhang 2019 (Ophiostomatales, Ophiostomataceae) (9 isolates) were also obtained from both the beetles and blue-stained tissue, as well as the isolates identified as *Graphilbum* cf. *rectangulosporium* Ohtaka, Masuya and Yamaoka (Ophiostomatales, Ophiostomataceae) (5 isolates). Isolates belonging to the species *Ophiostoma piceae* (Münch) Syd. and P. Syd. (Ophiostomatales, Ophiostomataceae) (6 isolates) and *Ophiostoma floccosum* Math–Käärik (Ophiostomatales, Ophiostomataceae) (3 isolates) were recovered only from the blue-stained tissues, and one isolate identified as *Ceratocystiopsis* cf. *minuta* (Siemaszko) H.P. Upadhyay and W.B. Kendr. (Ophiostomatales, Ophiostomataceae) was recovered from an adult *O. erosus* beetle. All the information about obtained and identified isolates can be found in Table 2. More details about the results of the blast analysis, including information about which species were identified by which barcode can be found in Appendix A.

## 4. Discussion

This is the first report of *Ophiostomatales* fungi identified as organisms associated with pest *O. erosus* and pine species in Croatia. Bark beetles are important forest pests, which have already been researched and discussed in relation to climate change, indicating that the predicted increase in temperature would lead to higher survival rates and faster development, thus directly influencing their population dynamics [4,30,51,52]. Recent observations of *O. erosus* outbreaks in the Dalmatian region of Croatia clearly indicate the onset of behavioural changes of this forest pest, previously considered benign in that area [31]. Drought intensity and frequency, and aridification trends in the research area caused cumulative stress to trees and have increased *O. erosus* occurrence. Population monitoring with pheromone traps in the years 2017, 2018, and 2019 showed up to 5–6 generations per year, a figure which has never been recorded in Croatia or in any other European country [31,53]. This study represents the first report of fungal species associated with *O. erosus* in Croatia, obtained from the adult beetles, their galleries and blue stained wood from *Pinus halepensis* and *Pinus pinea* at different locations in the Mediterranean part of Croatia. In previous studies, *O. erosus’s* association with blue-staining *Ophiostomatales* has already been demonstrated, and *Ophiostoma ips* was found to be the most common fungal symbiont of this bark beetle [34,54,55,56,57], which was the same as the results obtained in our research. In general, *O. ips* has been regarded as a common fungal species associated with blue stain and pine bark beetles in many countries such as Poland [58], Spain [59], Israel [60], Australia [61], South Africa [55], and China [62]. 

*Sporothrix pseudoabietina* was the second most isolated fungus, obtained both from the beetles and blue-stained tissue. In one study on *Ophiostomatales* associated with pines and pine bark beetles conducted in Australia *O. ips* and *S. pseudoabietina* were making up 53% and 19% of the dataset [58], which was a very similar share as in our study, with 50% isolates identified as *O. ips* and 15% as *S. pseudoabietina*. The genus *Sporothrix* has been recently redefined as a separate genus within the *Ophiostomatales* [60]. The species *S. pseudoabietina* was formally described in 2019, with the type specimen originating in China [42], and in Romon et al. [59] it was demonstrated as a commonly isolated fungus from *Pinus* spp. and accompanying bark beetles, and also in Jankowiak et al. [62] in association with spruce- and larch-infecting bark beetles.

*Ophiostoma piceae* was isolated only from blue-stained sapwood and accounted for 10% of the identified species. This species commonly causes wood-staining of coniferous timber and belongs to the most important blue-stain agents of pines in Europe [13,18,38]. Although *O. piceae* is generally not considered to be a strict associate of bark beetles and is often encountered in the absence of their activity [38], it was found in some studies to be associated with pine bark beetles and their galleries [33,60,62,63]. For example, in Romón et al. [60] it was identified as a fungus associated with *O. erosus* on *Pinus radiata* in Spain but in a smaller proportion than in our study (only 2%).

Associations of *G.* cf. *rectangulosporium* with pine bark beetles have previously been reported [55,57,59], and in several studies, this species was found in associations with *O. erosus*, but in different isolation frequencies. For example, in Romón et al. [59] the share of isolates that belonged to *G.* cf. *rectangulosporium* was 2.4%, and in Dori–Bachash et al. [60] the frequency of isolation was 24%. In our study, it accounted for 8% of the isolates and was recovered both from *O. erosus* and blue-stained tissue.

The species *Ophiostoma floccosum* is known to be a common associate of phloephagous bark beetles in Europe but is usually occurring at low frequencies [14,61,64]. In Romón et al. [60], it was associated with *O. erosus* at a proportion of 1%, and in the present study, it accounted for 5% of the isolates but was recovered only from blue-stained tissues. One isolate in our study was identified as *Ceratocystiopsis* cf. *minuta* and was obtained only from *O. erosus*. This fungus is associated with a wide variety of conifer-infesting bark beetles in many parts of the world i.e., [14,65,66].

Since the first discovery of symbiotic associations of conifer-infesting bark beetles with *Ophiostomatales* [66], a great number of studies on their interactions, species complexes, taxonomy, and their role in tree mortality have been conducted. Nevertheless, until today their precise role as symbionts still remains quite uncertain. There is no question that some tree-killing bark beetles have virulent fungal associations, and some of these fungi are undoubtedly capable of killing trees. There are few with high pathogenicity that survived the trees in the inoculation studies, unless unnaturally high levels of inoculum were used [3,65]. Many of these fungi are only recognized as the causal agents of blue stain (sap stain) in wood, which is an economical problem due to the lowering of the timber quality, which is especially recorded in pine plantations worldwide [19,58,66]. All sites in our study also consisted of monoculture dominated by *P. halepensis*.

This research has made a significant contribution to the knowledge of fungi transmitted by the pine bark beetle *O. erosus* and pines as its hosts and given that this issue has not been investigated in Croatia so far, the results provide a good basis and guidelines for further studies of their complex symbiotic relationship. Although our goal was to isolate and identify *Ophiostomatales* associated with *O. erosus* beetles, galleries, and blue-stained sapwood collected during beetle outbreaks that were never recorded before in Mediterranean Croatia, more in-depth systematic research is required to improve understanding of these fungus-vector associations, and to reveal greater diversity of this fungal community that probably exists. Future research should focus on climatic drivers, such as temperature and drought periods, that may be correlated with tree dieback, combined with pathogenicity studies that would explain the pathogenic capability of the fungi found. Addressing these questions would be valuable for forest pest management, as it could be used to estimate tree resistance and ultimately predict the risk of tree mortality in the case of a new or similar bark beetle outbreak.

## Figures and Tables

**Figure 1 jof-08-00788-f001:**
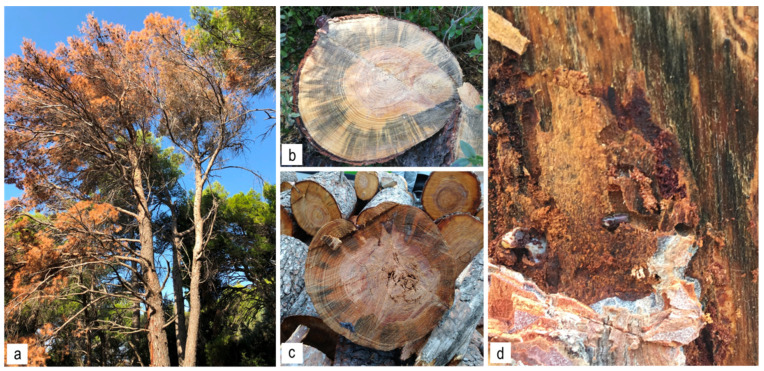
Disease symptoms observed on *Pinus halepensis* at Marjan Forest Park, Split. (**a**) Tree attacked by *Orthotomicus erosus*; (**b**,**c**) trunk sections with visible symptoms of blue staining in the sapwood; (**d**) galleries with *O. erosus* adults and visible blue staining.

**Figure 2 jof-08-00788-f002:**
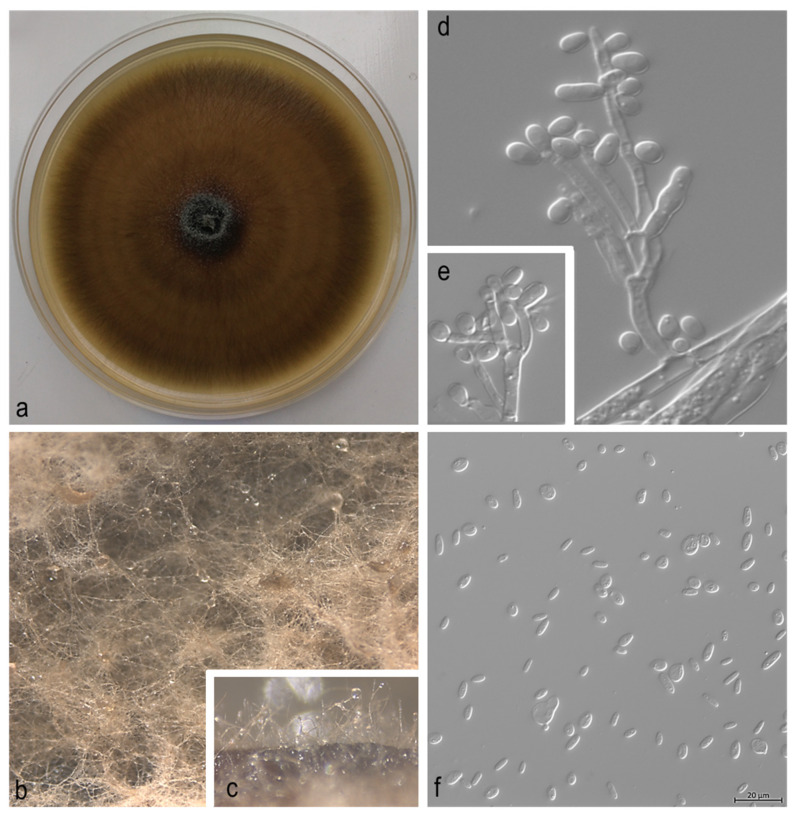
*Ophiostoma ips*. (**a**–**c**) Approximately 14-days old culture; (**d**,**e**) hyalorhinocladiella-like asexual morph: conidiogenous cells; (**f**) conidia.

**Figure 3 jof-08-00788-f003:**
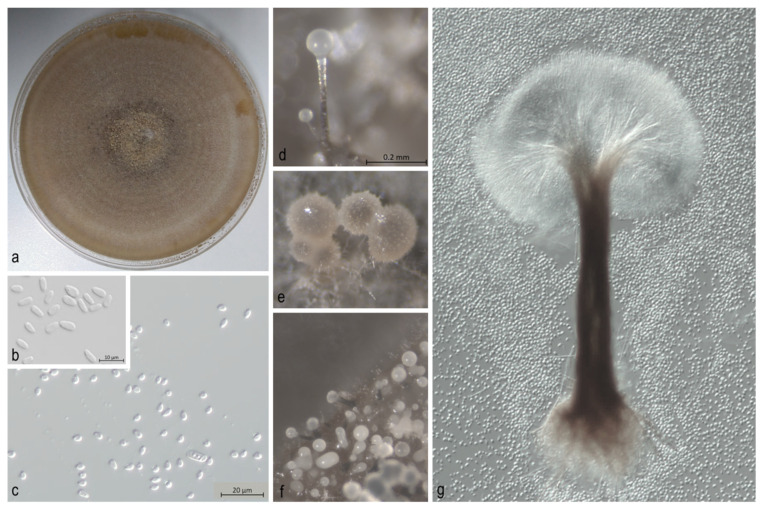
*Ophiostoma piceae*. (**a**) Approximately 14-days old culture; (**b**,**c**) conidia; (**d**–**f**) spore masses formed at the apices of fruiting structures; (**g**) pesotum-like asexual morph.

**Table 1 jof-08-00788-t001:** Year and number of samples collected regarding bark beetle infestation level and tree species.

Year	Forest Office	Management Unit	Tree Species	WGS Coordinates	Bark Beetle Infestation Level in 2018		
					Area [ha]	Wood Mass [m^3^]	Intensity [%]
2017 2018 2019	SPLIT	Marjan	*Pinus halepensis* *Pinus pinea*	43°30′33″ N, 16°24′38″ E	196	5.900	21–40
2018	KORČULA	Ošjak	*Pinus halepensis*	42°57′40″ N, 16°41′7″ E	18	1.800	21–40
2018	ZADAR	Sukošan	*Pinus halepensis*	44°02′54.0″ N, 15°19′22.1″ E	70	150	1–20
2019	DUBROVNIK	Lokrum	*Pinus halepensis*	42°37′38.7″ N, 18°07′15.9″ E	70	100	1–20

**Table 2 jof-08-00788-t002:** Information about obtained and identified isolates.

Isolate ID	Host Tree	Isolation Source	Year of Sampling	Locality	Identified Fungi
1OE1	*Pinus halepensis*	*Orthotomicus erosus*	2018	Marjan	*Ophiostoma ips*
2OE2	*Pinus halepensis*	*Orthotomicus erosus*	2018	Marjan	*Ophiostoma ips*
3OE3	*Pinus halepensis*	*Orthotomicus erosus*	2018	Marjan	*Ophiostoma ips*
4OE4	*Pinus halepensis*	*Orthotomicus erosus*	2018	Marjan	*Sporothrix* *pseudoabietina*
5OE5	*Pinus halepensis*	*Orthotomicus erosus*	2018	Marjan	*Pezizomycotina*
6OE6	*Pinus halepensis*	*Orthotomicus erosus*	2018	Marjan	*Ophiostoma ips*
7OE7	*Pinus halepensis*	*Orthotomicus erosus*	2018	Marjan	*Ceratocystiopsis* cf. *minuta*
8BS1	*Pinus halepensis*	blue stained wood	2018	Marjan	*Sporothrix pseudoabietina*
9OE8	*Pinus halepensis*	*Orthotomicus erosus*	2018	Marjan	*Sporothrix pseudoabietina*
10OE9	*Pinus pinea*	*Orthotomicus erosus*	2018	Marjan	*Ophiostoma ips*
11BS2	*Pinus pinea*	blue stained wood	2018	Marjan	*Ophiostoma ips*
12OE10	*Pinus halepensis*	*Orthotomicus erosus*	2018	Marjan	*Ophiostoma ips*
13OE11	*Pinus halepensis*	*Orthotomicus erosus*	2018	Marjan	*Ophiostoma ips*
14OE12	*Pinus halepensis*	*Orthotomicus erosus*	2018	Marjan	*Graphilbum* cf. *rectangulosporium*
15OE13	*Pinus halepensis*	*Orthotomicus erosus*	2018	Marjan	*Graphilbum* cf. *rectangulosporium*
16OE14	*Pinus halepensis*	*Orthotomicus erosus*	2018	Marjan	*Ophiostoma ips*
18OE15	*Pinus halepensis*	*Orthotomicus erosus*	2018	Marjan	*Sporothrix pseudoabietina*
19OE16	*Pinus halepensis*	*Orthotomicus erosus*	2018	Marjan	*Sporothrix* *pseudoabietina*
20BS3	*Pinus halepensis*	*blue stained wood*	2018	Ošjak	*Pezizomycotina*
20BS4	*Pinus halepensis*	blue stained wood	2018	Ošjak	*Pezizomycotina*
21BS5	*Pinus halepensis*	blue stained wood	2018	Ošjak	*Pezizomycotina*
22OE17	*Pinus halepensis*	*Orthotomicus erosus*	2018	Marjan	*Ophiostoma ips*
23OE18	*Pinus halepensis*	*Orthotomicus erosus*	2018	Marjan	*Ophiostoma ips*
24OE19	*Pinus halepensis*	*Orthotomicus erosus*	2018	Marjan	*Ophiostoma ips*
25OE20	*Pinus halepensis*	*Orthotomicus erosus*	2018	Marjan	*Ophiostoma ips*
26BS6	*Pinus pinea*	blue stained wood	2018	Marjan	*Ophiostoma ips*
27BS7	*Pinus halepensis*	blue stained wood	2018	Ošjak	*Pezizomycotina*
28OE21	*Pinus pinea*	*Orthotomicus erosus*	2018	Marjan	*Ophiostoma ips*
29G1	*Pinus pinea*	gallery	2018	Marjan	*Ophiostoma ips*
30G2	*Pinus pinea*	gallery	2018	Marjan	*Ophiostoma ips*
32OE22	*Pinus pinea*	*Orthotomicus erosus*	2018	Marjan	*Ophiostoma ips*
34OE23	*Pinus halepensis*	*Orthotomicus erosus*	2018	Marjan	*Ophiostoma ips*
35OE24	*Pinus halepensis*	*Orthotomicus erosus*	2018	Marjan	*Ophiostoma ips*
36OE25	*Pinus halepensis*	*Orthotomicus erosus*	2018	Marjan	*Sporothrix* *pseudoabietina*
37OE26	*Pinus halepensis*	*Orthotomicus erosus*	2018	Marjan	*Sporothrix* *pseudoabietina*
38OE27	*Pinus halepensis*	*Orthotomicus erosus*	2018	Marjan	*Sporothrix* *pseudoabietina*
39BS9	*Pinus halepensis*	blue stained wood	2019	Marjan	*Ophiostoma piceae*
40OE28	*Pinus pinea*	*Orthotomicus erosus*	2018	Marjan	*Ophiostoma ips*
41OE29	*Pinus halepensis*	*Orthotomicus erosus*	2018	Marjan	*Ophiostoma ips*
42BS10	*Pinus halepensis*	blue stained wood	2019	Marjan	*Graphilbum* cf. *rectangulosporium*
42BS11	*Pinus halepensis*	blue stained wood	2019	Marjan	*Graphilbum* cf. *rectangulosporium*
43OE30	*Pinus halepensis*	*Orthotomicus erosus*	2018	Marjan	*Ophiostoma ips*
44OE31	*Pinus halepensis*	*Orthotomicus erosus*	2018	Marjan	*Ophiostoma ips*
45OE32	*Pinus halepensis*	*Orthotomicus erosus*	2018	Marjan	*Ophiostoma ips*
46BS11	*Pinus pinea*	blue stained wood	2018	Marjan	*Ophiostoma ips*
47BS12	*Pinus pinea*	blue stained wood	2018	Marjan	*Ophiostoma ips*
48OE33	*Pinus halepensis*	*Orthotomicus erosus*	2018	Marjan	*Graphilbum* cf. *rectangulosporium*
49OE34	*Pinus halepensis*	*Orthotomicus erosus*	2018	Marjan	*Sporothrix* *pseudoabietina*
50BS13	*Pinus halepensis*	blue stained wood	2017	Marjan	*Ophiostoma ips*
51BS14	*Pinus halepensis*	blue stained wood	2017	Marjan	*Ophiostoma floccosum*
52BS15	*Pinus halepensis*	blue stained wood	2017	Marjan	*Ophiostoma floccosum*
53BS16	*Pinus halepensis*	blue stained wood	2017	Marjan	*Ophiostoma floccosum*
54BS17	*Pinus halepensis*	blue stained wood	2017	Marjan	*Ophiostoma piceae*
55BS18	*Pinus halepensis*	blue stained wood	2018	Sukošan	*Ophiostoma ips*
56BS19	*Pinus halepensis*	blue stained wood	2018	Sukošan	*Ophiostoma ips*
57BS20	*Pinus halepensis*	blue stained wood	2018	Sukošan	*Sarocladium strictum*
58BS21	*Pinus halepensis*	blue stained wood	2019	Lokrum	*Ophiostoma piceae*
59BS22	*Pinus halepensis*	blue stained wood	2019	Lokrum	*Ophiostoma piceae*
60BS23	*Pinus halepensis*	blue stained wood	2019	Lokrum	*Ophiostoma piceae*
61BS24	*Pinus halepensis*	blue stained wood	2019	Lokrum	*Ophiostoma piceae*

## Data Availability

Not applicable.

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
