# Peer review of "Ophiostomatales Associated with Mediterranean Pine Engraver, Orthotomicus erosus (Coleoptera, Curculionidae) in Dalmatia, Croatia"

_jof, 2022, doi:10.3390/jof8080788_

Round 1
Reviewer 1 Report
This is a well written, well conducted study that takes an important step towards bettering understanding of bark beetles, fungi and declines in this region. Good solid work, well worth publishing. May need a light edit for English usage, I found a few minor errors, there may be more.
A few minor corrections
Line 33 - Delete the word "and"
Line 37 - Change "tracheid" to "tracheids"
Line 40 - Change "associate" to "associates"
Line 198 - Need a literature reference to all this climate change - bark beetle literature you refer to. Also remember that much of that literature focuses as much on climate change effects on tree hosts as it does on beetles.
Author Response
This is a well written, well conducted study that takes an important step towards bettering understanding of bark beetles, fungi and declines in this region. Good solid work, well worth publishing. May need a light edit for English usage, I found a few minor errors, there may be more.
A few minor corrections
Line 33 - Delete the word "and"
Done
Line 37 - Change "tracheid" to "tracheids"
Done
Line 40 - Change "associate" to "associates"
Done
Line 198 - Need a literature reference to all this climate change - bark beetle literature you refer to. Also remember that much of that literature focuses as much on climate change effects on tree hosts as it does on beetles.
Thank you for this suggestion: We add this publications:
- Caroll AL, Taylor SW, Regniere J, Safranyik 2004 Effects on climate change on range expansion by the mountain pine beetle in British Columbia. In: Shore TL, Brooks JE, Stone JE (eds) Mountain Pine Beetle Symposium: Challenges and Solutions, Information Report BC-X-399, Kelowna, British Columbia, Canada, 30-31 October 2003. Canadian Forest Service, Pacific Forestry Centre, Victoria, British Columbia, Canada, pp 223-232
- Pureswaran DS, Roques A, Battisti A 2018 Forest insects and climate change. Curr Forestry Rep 4 (2): 35-50. DOI: https://doi.org/10.1007/s40725-018-0075-6

Reviewer 2 Report
The authors reported six ophiostomatoid fungal species associated with Orthotomicus erosus in Croatia, and claimed that their research has made a great contribution to the knowledge of fungi transmitted by the pine bark beetle O. erosus and pines as its hosts. The results provide a good basis and guidelines for further studies of their complex symbiotic relationship.
The major problem in the manuscript is lacking of phylogenetic analysis, which suggests that the authors may not be very knowledgeable about the taxonomy of ophiostomatoid fungi. The authors stated that the main purpose of this study was to isolate and identify fungi associated with O. erosus, and elucidate their relationship with this beetle. However, the identification results in this paper are questionable and the relationship between beetles and fungi is not carefully compared, so I do not think this paper should be published until these issues are resolved.
Other specific comments:
The term “ophiostomatoid fungi” is used throughout, but there is no mention of Microascales species, so I think it should be replaced with “Ophiostomatales”.
Line 38-40: The order Ophiostomatales has been revised by de Beer et al. (2022)
doi: 10.3114/sim.2022.101.02
Line 99: Two weeks are too long to isolate fungi because some fungi can grow even at 4℃.
Line 224-225: Pinus and Ophiostoma piceae should be in italic
Figure: The morphological characteristics of O. ips are rough. It is easy to observe the perithecium with very long neck of this species, but the authors do not show it in the manuscript. In addition, the authors claim to have identified six species, but why only two are shown (Figure 2 and Figure 3). What puzzled me was that the authors showed the morphological diagrams of the two species, but did not describe the morphological characteristics.
Table 1: According to “one fungus, one name”, “Ophiostoma rectangulosporium” should no longer be used here.
Table S1: I have great concern that the result of fungal identification. It is unprofessional and unreliable to rely solely on blast results. In addition, the title of Table S1 was incorrectly written as Table 2.
Author Response
The authors reported six ophiostomatoid fungal species associated with Orthotomicus erosus in Croatia, and claimed that their research has made a great contribution to the knowledge of fungi transmitted by the pine bark beetle O. erosus and pines as its hosts. The results provide a good basis and guidelines for further studies of their complex symbiotic relationship.
The major problem in the manuscript is lacking of phylogenetic analysis, which suggests that the authors may not be very knowledgeable about the taxonomy of ophiostomatoid fungi. The authors stated that the main purpose of this study was to isolate and identify fungi associated with O. erosus, and elucidate their relationship with this beetle. However, the identification results in this paper are questionable and the relationship between beetles and fungi is not carefully compared, so I do not think this paper should be published until these issues are resolved.
Other specific comments:
The term “ophiostomatoid fungi” is used throughout, but there is no mention of Microascales species, so I think it should be replaced with “Ophiostomatales”.
We agree. Done throughout the text.
Line 38-40: The order Ophiostomatales has been revised by de Beer et al. (2022)
doi: 10.3114/sim.2022.101.02
Line 99: Two weeks are too long to isolate fungi because some fungi can grow even at 4℃.
You are right, that would be too long. The isolation was performed within few days after collection of samples. The text refers to the all period of isolation that took 2 weeks for the bark beetles, their galleries and stained sapwood. For example, one day isolation was performed from the bark beetles that were took from the bark at the trunk sections stored in laboratory, the other day it was performed from the sapwood that was also cut from the same trunk sections etc. Also, as soon as the mycelium started to grow from the samples inoculated in nutrient medium it was transferred to fresh MEA or PDA medium so the pure cultures can be obtained from all the developed fungi. All this process lasted about two weeks.
We add a sentence to clarify that:
The isolation of fungi was performed within few day after collection of samples.
Line 224-225: Pinus and Ophiostoma piceae should be in italic
Done
Figure: The morphological characteristics of O. ips are rough. It is easy to observe the perithecium with very long neck of this species, but the authors do not show it in the manuscript. In addition, the authors claim to have identified six species, but why only two are shown (Figure 2 and Figure 3). What puzzled me was that the authors showed the morphological diagrams of the two species, but did not describe the morphological characteristics.
Unfortunately, we did not take good quality pictures of the other species so we decided to use the ones we had (those two are also the most common species found in this research). In general, we focused more on the association of important pest Orthotomicus erosus with Ophiostomatales, since we have a serious decline of pines in Croatian coast, and now it seems that the problem is more complex as we previously thought. This research is just a piece of a puzzle and we are aware that more in depth research is needed.
Table 1: According to “one fungus, one name”, “Ophiostoma rectangulosporium” should no longer be used here.
We changed it to "Graphilbum rectangulosporium"
Table S1: I have great concern that the result of fungal identification. It is unprofessional and unreliable to rely solely on blast results. In addition, the title of Table S1 was incorrectly written as Table 2.
Thank you, we corrected the title of the table. Unfortunately, in this phase we are not able to perform phylogenetic analysis so we need to rely only on blast results.
Reviewer 3 Report
Heidy Luo
Assistant Editor
Journal of Fungi
The review of the manuscript entitled “Ophiostomatoid fungi associated with Mediterranean pine engraver, Orthotomicus erosus (Coleoptera, Curculionidae)” (jof-1794637).
This paper provides results of a survey of the Orthotomicus erosus-associated fungi in Croatia. The scientific concept is well thought out. Moreover, I’d like to highlight some aspects regarding the importance of evaluated research: 1) to date, no studies in Croatia (and generally in the Mediterranean region) on bark beetles-associated fungi have been done; 2) relatively little is known regarding the biology of ophiostomatoid fungi associated with Orthotomicus erosus, 3) the health condition of Pinus spp. in Dalmatia is wrong and knowing all factors affect tree health are therefore important to consider.
This manuscript has only a few shortcomings, including one serious problem. I recommend this manuscript to Journal of Fungi after major revision.
Major remark
Although sequences of ITS, TUB2 and TEF1 for representative isolates of the ophiostomatoid have been obtained in this I have doubts about correct fungal identification. Recently the phylogenetic analysis of the genera or species complexes within Ophiostoma s. lato and Leptographium s. lato have become the norm. Therefore I recommend adding the phylogenetic trees for Ophiostoma, Sporothrix and Graphilbum. Ideally, this would be combined trees or trees based on ITS and TUB2 although.
The manuscript is poorly written, includes many errors and should be re-written in many places. My remarks are below:
Tittle
I suggest the following title:
Ophiostomatoid fungi (Ascomycota) associated with Mediterranean pine engraver, Orthotomicus erosus (Coleoptera, Curculionidae) in Dalmatian, Croatia.”
Introduction
Line 29 for example [4-6]….? It looks weird. Please give the examples and later add the references [4-6].The same applies [7-8].
Line 30-31 ‘It is assumed….”. The paradigm that phytopathogenic fungi associated with tree-killing bark beetles are critical for overwhelming tree defenses and incurring host tree mortality has been contested by Six and Wingfield in 2011 (Six DL, Wingfield MJ. The role of phytopathogenicity in bark beetle-fungus symbioses: a challenge to the classic paradigm. Annu Rev Entomol. 2011;56:255-72. doi: 10.1146/annurev-ento-120709-144839). Authors suggested that phytopathogenicity is rather important for the fungi not for beetle species. Please note it in the text.
Line 38-39 Please change this sentence to:
They belong to the ascomycete families Ceratocystidaceae and Graphiaceae (order Microascales) and the singly family Ophiostomataceae in the order Ophiostomatales (Hyde et al. 2020).
Hyde KD, Norphanphoun C, Maharachchikumbura SSN, Bhat DJ, Jones EBG, Bundhun D, Chen YJ, Bao DF, Boonmee S, Calabon MS, et al. 2020. Refined families of Sordariomycetes. Mycosphere 11(1):305–1059. doi:10.5943/mycosphere/11/1/7
I think that authors should give information about O. erosus-associated fungi based on surveys from RSA(Zhou et al. 2001), Spain (Ramon et al. 2007, Malacrino et al. 2017), Israeal (Dori-Bachash et al. 2015), Tunisia (Jamaa et al. 2007) and Morocco (de Beer et al. 2004).
Materials and methods
1) Section of study sites and collection samples needs complementing:
· Samples were collected in 2017-2019 from four pine stands in Dalmatia. However, we have no detailed information about sampling sites (e.g. characterization of pine stands (age, area), level of infestation by O. erosus (maybe other beetle species also?) and health status of trees infested by this beetle species (dead, dying?). How many samples were collected from each study site?
· I expect more information about collected samples, e.g. how many trees and samples (beetles and galleries) did take to fungal isolation? In what way galleries were collected from infested trees (trees were cut?). What kind of beetles were taken to the fungal isolation: adults boring into tree during flight period or maybe young adults before leave the tree. Please provide information when samples were taken (spring, summer, autumn)?
2) Section of identification of fungi and DNA extraction….
Information about procedures of the molecular identification of representative strains has not been included. Because the authors didn’t perform the phylogenetic analysis we need more information about procedures of fungal identification.
For example:
The sequences of ITS regions and protein-coding genes were compared with sequences in GenBank using BLAST. Isolates with sequence similarity to the ex-type strains exceeding …..?% were considered conspecific.
However, I consider that the phylogenetic analysis based on ITS, TUB2 and TEF1 sequences are needed.
I am not sure if Graphilbum rectangulisporium (=Ophiostoma rectangulosporium) and Ceartocystiopsis minuta are correctly identified.
Results
1) Line 161: …revealed seven…” Please change to ….revealed six species belonging to the Ophiostomatales. Next sentence should be delete. There is no need to repeat (Ophiostomatales, Ophiostomataceae) later in the text.
2) Line 168 Does Journal of Fungi require you to unitalicize phylum or orders, family names? If not, we should write Ophiostomatales and Ophiostoamataceae in italicized format. I would point out that all higher levels (family, order, etc) should be in italics as per recommendation of the ICTF.
3) Lines 169-179 “…which was recovered from both beetles (70%) and blue-stained tissue/galleries (30%)”. What means (70%) or (30%). Is this the isolation frequency? What it was calculated? Later, (lines 173-179) authors did not provide these values.
I think that authors should create the table tilted: “Frequencies (%) of ophiostomatoid fungi obtained from Orthotomicus erosus beetles (B) and their galleries (G) collected from pines in Croatia” with the breakdown for each study site and beetles and galleries.
Were there differences between P. halepensis nad P. pinea?
4) Lines 170 and 191. Figures 2 and 3. Ophiostoma ips and Ophiostoma piceae are very well-known species and I see no need to placing these pictures in the manuscript.
5) Line 180 Ceartocystiopsis minuta species complex includes many phylogenetic similar species. Recently Jankowiak et al. 2022 (Jankowiak R., Solheim H., Bilański P., Mukhopadhyay J., Hausner G. 2022. Ceratocystiopsis spp. associated with pine- and spruce-infesting bark beetles in Norway. Mycological Progress 21, 61) described four new Ceratocystiopsis species closely related to C. minuta s. stricto (isolate UAMH 11218 (= a dried culture of UM1532 = WIN(M)1532), originally isolated from P. abies infested by I. typographus in the Biebrzański National Park (Poland). I am not sure if isolate 7OE7 represents C. minuta s. stricto.
Line 175 I'm having second thoughts about Ophiostoma rectangulosporium. The correct name is Graphilbum rectangulisporium. This species is known only from bark beetle-infested Abies species in Japan (Ohtaka et al. 2006). I suppose that the Croatian isolates may represents other species (e.g. G. crescericum). Therefore the phylogenetic analysis are needed (see paper Jankowiak R., Solheim H., Bilański P., Marincowitz S., Wingfield M.J. 2020. Seven new species of Graphilbum from conifers in Norway, Poland, and Russia. Mycologia 112(6): 1240–1262.).
Incidentally, in legend of figure 2 should (c) and (d) hyalorhinocladiella-like asexual morph: conidiogenous cells. In addition, in my opinion picture e shows synnemata of Ophiostoma picaea. Ophiostoma ips does not produce pesotum-like asexual morph in culture.
In figure 3 should be: (g) pesotum-like asexual morph
Lines 182-185: Please delete this sentence. Sarocladium strictum does not belong to the ophiostomatoid fungi.
Lines 189-190.
I think that with this sentence, the discussion should be begin.
Discussion
1) Lines 208-209…”and results showed that 92% % of all identified species belonged to the ophiostomatoid group of fungi”. Please it delete.
2) Lines 214-215. Change Australia to Spain references [57] for China should be change to:
Chang R, Duong TA, Taerum SJ, Wingfield MJ, Zhou X, de Beer ZW (2017) Ophiostomatoid fungi associated with conifer-infesting beetles and their phoretic mites in Yunnan, China. MycoKeys 28: 19-64. https://doi.org/10.3897/mycokeys.28.21758
3) Line 215. Malacrino et al. 2017 probably also found O. ips in association with O. erosus. Please check.
Malacrino et al. 2017. Fungal communities associated with bark and ambrosia beetles
trapped at international harbours. Fungal biology 28: 44-52
Lines 216-220. I don’t understand. The paper [56] is about bark beetles associated with Ophiostomatales including O. erosus in Spain not in Australia. The authors write here about other paper! probably about Trollip et al. 2021. Moreover Trollip et al. 2021 studied fungal associates of Ips grandicollis, Hylurgus ligniperda and Hylastes ater on P. radiata and P. ponderosae, P. elliottii P. caribaea x elliottii not O. erosus!
Line 221 please change (57) to [57]
Line 223 Reference [56] is Romon not Trollip.
Lines 221-224 Sporothrix pseudoabietina has been found also by Jankowiak et al. 2017 (Jankowiak R., Strzałka B., Bilański P., Kacprzyk M., Lukášová K., Linnakoski R., Matwiejczuk S., Misztela M., Rossa R. 2017. Diversity of Ophiostomatales species associated with conifer infesting beetles in the Western Carpathians. European Journal of Forest Research 136: 939–956) as Sporothrix sp. 1. in association with spruce- and larch-infecting bark beetles.
Line 226 In section results is not given that O. piceae had 10% of frequency.
Lines 225 Ophiostoma piceae in italics
Lines 230-231 Please change 55 to 56
Lines 233-238 About G. rectangulisporium I wrote above. Isolates called as Ophiostoma rectangulosporium-like in paper Romon et al. 2007 have been described later as G. crescericum in paper Romon et al. 2014.
Is not clear which Graphilbum species is Croatian.
Lines 240-241 I did not find O. floccosum in paper [54]. Instead of Ben Jamaa paper you can cited (Jankowiak et al. 2017).
Lines 243-246. About Ceratocystiopsis minuta I wrote above.
Lines 247-251 Please delete this paragraph.
Lines 252-263. I think that this paragraph is not needed and is not associated with this work. Instead of this, please compare the fungal community structure of O. erosus obtained in this study with other studies (Zhou et al. 2001; Romon et al. 2007; Dori-Bachash et al.2015). For example, O. erosus was associated with seven taxa in Spain and only O. rectangulosporium-like has been also identified in Croatia.
What about Geosmithia spp.? According to Dori-Bachash et al. (2015), O. erosus is commonly associated with these fungi in Israel.
References
Line 358 Reference [34] is not citied in the text.
Author Response
The review of the manuscript entitled “Ophiostomatoid fungi associated with Mediterranean pine engraver, Orthotomicus erosus (Coleoptera, Curculionidae)” (jof-1794637).
This paper provides results of a survey of the Orthotomicus erosus-associated fungi in Croatia. The scientific concept is well thought out. Moreover, I’d like to highlight some aspects regarding the importance of evaluated research: 1) to date, no studies in Croatia (and generally in the Mediterranean region) on bark beetles-associated fungi have been done; 2) relatively little is known regarding the biology of ophiostomatoid fungi associated with Orthotomicus erosus, 3) the health condition of Pinus spp. in Dalmatia is wrong and knowing all factors affect tree health are therefore important to consider.
This manuscript has only a few shortcomings, including one serious problem. I recommend this manuscript to Journal of Fungi after major revision.
Major remark
Although sequences of ITS, TUB2 and TEF1 for representative isolates of the ophiostomatoid have been obtained in this I have doubts about correct fungal identification. Recently the phylogenetic analysis of the genera or species complexes within Ophiostoma s. lato and Leptographium s. lato have become the norm. Therefore I recommend adding the phylogenetic trees for Ophiostoma, Sporothrix and Graphilbum. Ideally, this would be combined trees or trees based on ITS and TUB2 although.
The manuscript is poorly written, includes many errors and should be re-written in many places. My remarks are below:
Tittle
I suggest the following title:
Ophiostomatoid fungi (Ascomycota) associated with Mediterranean pine engraver, Orthotomicus erosus (Coleoptera, Curculionidae) in Dalmatian, Croatia.”
Thank you, considering also the other reviewer we changed the title to:
Ophiostomatales associated with Mediterranean pine engraver, Orthotomicus erosus (Coleoptera, Curculionidae) in Dalmatia, Croatia
Introduction
Line 29 for example [4-6]….? It looks weird. Please give the examples and later add the references [4-6].The same applies [7-8].
We drop “for example”
Line 30-31 ‘It is assumed….”. The paradigm that phytopathogenic fungi associated with tree-killing bark beetles are critical for overwhelming tree defenses and incurring host tree mortality has been contested by Six and Wingfield in 2011 (Six DL, Wingfield MJ. The role of phytopathogenicity in bark beetle-fungus symbioses: a challenge to the classic paradigm. Annu Rev Entomol. 2011;56:255-72. doi: 10.1146/annurev-ento-120709-144839). Authors suggested that phytopathogenicity is rather important for the fungi not for beetle species. Please note it in the text.
Thank you for this suggestion. We add this publication.
Line 38-39 Please change this sentence to:
They belong to the ascomycete families Ceratocystidaceae and Graphiaceae (order Microascales) and the singly family Ophiostomataceae in the order Ophiostomatales (Hyde et al. 2020).
Thank you for this suggestion. We add this publication.
Hyde KD, Norphanphoun C, Maharachchikumbura SSN, Bhat DJ, Jones EBG, Bundhun D, Chen YJ, Bao DF, Boonmee S, Calabon MS, et al. 2020. Refined families of Sordariomycetes. Mycosphere 11(1):305–1059. doi:10.5943/mycosphere/11/1/7
I think that authors should give information about O. erosus-associated fungi based on surveys from RSA(Zhou et al. 2001), Spain (Ramon et al. 2007, Malacrino et al. 2017), Israeal (Dori-Bachash et al. 2015), Tunisia (Jamaa et al. 2007) and Morocco (de Beer et al. 2004).
Materials and methods
1) Section of study sites and collection samples needs complementing:
- Samples were collected in 2017-2019 from four pine stands in Dalmatia. However, we have no detailed information about sampling sites (e.g. characterization of pine stands (age, area), level of infestation by O. erosus (maybe other beetle species also?) and health status of trees infested by this beetle species (dead, dying?). How many samples were collected from each study site?
- I expect more information about collected samples, e.g. how many trees and samples (beetles and galleries) did take to fungal isolation? In what way galleries were collected from infested trees (trees were cut?). What kind of beetles were taken to the fungal isolation: adults boring into tree during flight period or maybe young adults before leave the tree. Please provide information when samples were taken (spring, summer, autumn)?
We add: For sampling only wilting and drying pine trees clearly attacked by O.erosus were collected. On every collection site another bark beetle species Tomicus destruens Woll. could be found but in very low population. No tree was attacked by both bark beetle species at the same time.
We also add new Table 1 for clarification.
Table 1. Year and number of samples collected regarding bark beetle infestation level and tree species
Year |
Forest Office |
Management unit |
Tree species |
WGS coordinates |
Bark beetle infestation level in 2018 |
||
Area [ha] |
Wood mass [m3] |
Intensity [%] |
|||||
2017 2018 2019 |
SPLIT |
Marjan |
Pinus halepensis Pinus pinea |
43°30´33˝N, 16°24´38˝E |
196 |
5.900 |
21-40 |
2018 |
KORČULA |
Ošjak |
Pinus halepensis |
42°57´40˝N, 16°41´7˝E |
18 |
1.800 |
21-40 |
2018
|
ZADAR |
Sukošan |
Pinus halepensis |
44°02'54.0"N, 15°19'22.1"E |
70 |
150 |
1-20 |
2019 |
DUBROVNIK |
Lokrum |
Pinus halepensis |
42°37'38.7"N, 18°07'15.9"E |
70 |
100 |
1-20 |
2) Section of identification of fungi and DNA extraction….
Information about procedures of the molecular identification of representative strains has not been included. Because the authors didn’t perform the phylogenetic analysis we need more information about procedures of fungal identification.
For example:
The sequences of ITS regions and protein-coding genes were compared with sequences in GenBank using BLAST. Isolates with sequence similarity to the ex-type strains exceeding …..?% were considered conspecific.
However, I consider that the phylogenetic analysis based on ITS, TUB2 and TEF1 sequences are needed.
I am not sure if Graphilbum rectangulisporium (=Ophiostoma rectangulosporium) and Ceartocystiopsis minuta are correctly identified.
We are sure for the G. rectangulisporium but in previous works from Europe (e.g. Jankowiak 2012) it was written as cf. C. minuta had a bit poor sequence. Because of that we decided to use confer in the text for both species.
Results
1) Line 161: …revealed seven…” Please change to ….revealed six species belonging to the Ophiostomatales. Next sentence should be delete. There is no need to repeat (Ophiostomatales, Ophiostomataceae) later in the text.
Done.
2) Line 168 Does Journal of Fungi require you to unitalicize phylum or orders, family names? If not, we should write Ophiostomatales and Ophiostoamataceae in italicized format. I would point out that all higher levels (family, order, etc) should be in italics as per recommendation of the ICTF.
Done.
3) Lines 169-179 “…which was recovered from both beetles (70%) and blue-stained tissue/galleries (30%)”. What means (70%) or (30%). Is this the isolation frequency? What it was calculated? Later, (lines 173-179) authors did not provide these values.
It means 70% were isolated from beetles and 30% from tissue, but it seems confusing so we drop that.
I think that authors should create the table tilted: “Frequencies (%) of ophiostomatoid fungi obtained from Orthotomicus erosus beetles (B) and their galleries (G) collected from pines in Croatia” with the breakdown for each study site and beetles and galleries.
We think that the data in the Table 2. are covering that.
Were there differences between P. halepensis nad P. pinea?
There were no differences between pine host species.
4) Lines 170 and 191. Figures 2 and 3. Ophiostoma ips and Ophiostoma piceae are very well-known species and I see no need to placing these pictures in the manuscript.
We would like to keep that because of the readers/professionals from Croatia.
5) Line 180 Ceartocystiopsis minuta species complex includes many phylogenetic similar species. Recently Jankowiak et al. 2022 (Jankowiak R., Solheim H., Bilański P., Mukhopadhyay J., Hausner G. 2022. Ceratocystiopsis spp. associated with pine- and spruce-infesting bark beetles in Norway. Mycological Progress 21, 61) described four new Ceratocystiopsis species closely related to C. minuta s. stricto (isolate UAMH 11218 (= a dried culture of UM1532 = WIN(M)1532), originally isolated from P. abies infested by I. typographus in the Biebrzański National Park (Poland). I am not sure if isolate 7OE7 represents C. minuta s. stricto.
We propose as mentioned above to use confer in the species name (C. cf. minuta)
Line 175 I'm having second thoughts about Ophiostoma rectangulosporium. The correct name is Graphilbum rectangulisporium. This species is known only from bark beetle-infested Abies species in Japan (Ohtaka et al. 2006). I suppose that the Croatian isolates may represents other species (e.g. G. crescericum). Therefore the phylogenetic analysis are needed (see paper Jankowiak R., Solheim H., Bilański P., Marincowitz S., Wingfield M.J. 2020. Seven new species of Graphilbum from conifers in Norway, Poland, and Russia. Mycologia 112(6): 1240–1262.).
Yes, we are aware of that problem. We propose to use confer for this species (Graphilbum cf. rectangulisporium)
Incidentally, in legend of figure 2 should (c) and (d) hyalorhinocladiella-like asexual morph: conidiogenous cells. In addition, in my opinion picture e shows synnemata of Ophiostoma picaea. Ophiostoma ips does not produce pesotum-like asexual morph in culture.
Thank you. We replaced this part of figure with other pictures.
In figure 3 should be: (g) pesotum-like asexual morph
Done.
Lines 182-185: Please delete this sentence. Sarocladium strictum does not belong to the ophiostomatoid fungi.
Done.
Lines 189-190.
I think that with this sentence, the discussion should be begin.
Done.
Discussion
- Lines 208-209…”and results showed that 92% % of all identified species belonged to the ophiostomatoid group of fungi”. Please it delete.
Done
2) Lines 214-215. Change Australia to Spain references [57] for China should be change to:
Chang R, Duong TA, Taerum SJ, Wingfield MJ, Zhou X, de Beer ZW (2017) Ophiostomatoid fungi associated with conifer-infesting beetles and their phoretic mites in Yunnan, China. MycoKeys 28: 19-64. https://doi.org/10.3897/mycokeys.28.21758
3) Line 215. Malacrino et al. 2017 probably also found O. ips in association with O. erosus. Please check.
Malacrino et al. 2017. Fungal communities associated with bark and ambrosia beetles
trapped at international harbours. Fungal biology 28: 44-52
You are right, we delete that (Our study is the first report of O. ips on O. erosus in Europe).
Lines 216-220. I don’t understand. The paper [56] is about bark beetles associated with Ophiostomatales including O. erosus in Spain not in Australia. The authors write here about other paper! probably about Trollip et al. 2021. Moreover Trollip et al. 2021 studied fungal associates of Ips grandicollis, Hylurgus ligniperda and Hylastes ater on P. radiata and P. ponderosae, P. elliottii P. caribaea x elliottii not O. erosus!
Yes you are right. It is Trollip and we changed that.
Line 221 please change (57) to [57]
Done
Line 223 Reference [56] is Romon not Trollip.
Yes, you are right. It is Romon and we changed that.
Lines 221-224 Sporothrix pseudoabietina has been found also by Jankowiak et al. 2017 (Jankowiak R., Strzałka B., Bilański P., Kacprzyk M., Lukášová K., Linnakoski R., Matwiejczuk S., Misztela M., Rossa R. 2017. Diversity of Ophiostomatales species associated with conifer infesting beetles in the Western Carpathians. European Journal of Forest Research 136: 939–956) as Sporothrix sp. 1. in association with spruce- and larch-infecting bark beetles.
Thank you. We add this citation.
Line 226 In section results is not given that O. piceae had 10% of frequency.
Lines 225 Ophiostoma piceae in italics
Done.
Lines 230-231 Please change 55 to 56
Done
Lines 233-238 About G. rectangulisporium I wrote above. Isolates called as Ophiostoma rectangulosporium-like in paper Romon et al. 2007 have been described later as G. crescericum in paper Romon et al. 2014.
Is not clear which Graphilbum species is Croatian.
We propose confer as mentioned above.
Lines 240-241 I did not find O. floccosum in paper [54]. Instead of Ben Jamaa paper you can cited (Jankowiak et al. 2017).
Done
Lines 243-246. About Ceratocystiopsis minuta I wrote above.
Explanation above.
Lines 247-251 Please delete this paragraph.
Done.
Lines 252-263. I think that this paragraph is not needed and is not associated with this work. Instead of this, please compare the fungal community structure of O. erosus obtained in this study with other studies (Zhou et al. 2001; Romon et al. 2007; Dori-Bachash et al.2015). For example, O. erosus was associated with seven taxa in Spain and only O. rectangulosporium-like has been also identified in Croatia.
What about Geosmithia spp.? According to Dori-Bachash et al. (2015), O. erosus is commonly associated with these fungi in Israel.
It was not found in our studies.
References
Line 358 Reference [34] is not citied in the text.
It is now corrected.

Reviewer 4 Report
This is interesting, well written paper, based on sound methodology and presenting novel data.
Comments;
1) the most important: in the main text, exchange Table 1 with the Supplementary Table, and, vice versa, present Table 1 as the Supplementary Table; in the revised version refer to those accordingly (e.g., see comment 6);
2) line 33: remove “and”;
3) line 40: “associates”;
4) lines 230-231: “For example, in Romón et al. [55] it was …”; but in the References this source is listed as no. 56 (line 405); therefore, check numbering of all citations and the reference list thoroughly;
5) lines 78-82, presenting coordinates of the localities; in this respect, a map showing Croatia in the European context and the localities within the country would be informative for a wide international readership (in this case as Figure 1, and then renumbering following figures accordingly);
6) lines 128-129 would imply the following questions: how many isolates from insects, from galleries and from wood? how many morphotypes defined in all? how many of those morphotypes were represented by the 60 isolates subjected to molecular studies? the answers are mainly reflected in the tables, just refer to those in the text briefly;
7) line 225: “Ophiostoma piceae” in Latin lettering;
8) lines 252-262: this paragraph to a certain extent in mirrors the first paragraph; moreover, studies on pathogenicity was not part of the study; could be removed;
9) line 264: exchange “great” with “significant”.
Author Response
This is interesting, well written paper, based on sound methodology and presenting novel data.
Comments;
- the most important: in the main text, exchange Table 1 with the Supplementary Table, and, vice versa, present Table 1 as the Supplementary Table; in the revised version refer to those accordingly (e.g., see comment 6);
According to other reviewers we would like to keep it that way.
- line 33: remove “and”;
Done
- line 40: “associates”;
Done
- lines 230-231: “For example, in Romón et al. [55] it was …”; but in the References this source is listed as no. 56 (line 405); therefore, check numbering of all citations and the reference list thoroughly;
Thank you, we checked it and it is now corrected.
5) lines 78-82, presenting coordinates of the localities; in this respect, a map showing Croatia in the European context and the localities within the country would be informative for a wide international readership (in this case as Figure 1, and then renumbering following figures accordingly);
We agree with you, but there is not enough space because we need to add a new Table 1 according to other reviewers’ requests. We believe that in that way it shall be informative.
- lines 128-129 would imply the following questions: how many isolates from insects, from galleries and from wood? how many morphotypes defined in all? how many of those morphotypes were represented by the 60 isolates subjected to molecular studies? the answers are mainly reflected in the tables, just refer to those in the text briefly;
Thank you. We add the linked table.
7) line 225: “Ophiostoma piceae” in Latin lettering;
Done
8) lines 252-262: this paragraph to a certain extent in mirrors the first paragraph; moreover, studies on pathogenicity was not part of the study; could be removed;
We removed that.
9) line 264: exchange “great” with “significant”.
Done.

Round 2
Reviewer 2 Report
Despite the improved quality of the revised manuscript, phylogenetic analysis has not been added, which is fatal. Therefore, I still don't think this article can be published on JOF
Reviewer 3 Report
Most of my comments have been included in the corrected manuscript. Therefore I recommend the manuscript in present form in Journal of Fungi.